# IMPACT OF PROMPT ON LATENT REPRESENTATIONS IN LLMS

## ABSTRACT

The effectiveness of zero-shot learning frameworks, particularly in Large Language Models (LLMs), has lately shown tremendous improvement. Nonetheless, zero-shot performance critically depends on the prompt quality. Scientific literature has been prolific in proposing methods to select, create, and evaluate prompts from a language or performance perspective, changing their phrasing or creating them following heuristics rules. While these approaches are intuitive, they are insufficient in unveiling the internal mechanisms of Large Language Models. In this work, we propose exploring the impact of prompts on the latent representations of auto-regressive transformer models considering a zero-shot setting. We focus on the geometrical properties of prompts' inner representation at different stages of the model. Experiments conducted give insights into how prompt characteristics influence the structure and distribution of vector representations in generative models. We focus on binary classification tasks on which prompting methods have shown robust performance and show that prompt formulation has indeed an influence on latent representation. However, their impact is dependent on the model family. Using clustering methods, we show that even though prompts are similar in natural language, surprisingly, their representations can differ. This is highly model-dependent, demonstrating the need for more precise analysis.

## 1 INTRODUCTION

It has recently been demonstrated that language models are capable of scaling to billions of parameters, achieving unprecedented performance on a range of natural language processing tasks (Brown et al., 2020; Hu et al., 2022). This novel parameter scale can be attributed to two key factors. Firstly, most of these models are based on the transformer architecture (Vaswani et al., 2017), which allows for straightforward parallelization and thus uses more computing power. Secondly, they all employ the pre-training paradigm, making them a robust transfer learning tool (Devlin et al., 2019; Radford et al., 2018). However, the sheer number of parameters comes with a significant drawback. The process of tuning a model is not cost- nor energy-efficient (Wang et al., 2023; Luccioni et al., 2023).

Thankfully, an unexpected phenomenon emerged from the hundred million parameter scale: robust few-shot learning (Brown et al., 2020), which can be approached in a no-training fashion named in-context-learning. In-context Learning can simply be rephrased: The model learns what it is supposed to do using its given input (Brown et al., 2020; Raffel et al., 2020). More precisely, modifying inputs accordingly to the desired downstream tasks (*i.e.* giving some example or describing the task) gives satisfying results. The zero-shot setting is an even more impressive achievement which is observed at the billion of parameters scale. Giving some examples to the context is not necessary anymore. A precise description of the task can, indeed, produce good results on a new task (Wei et al., 2022).
Both settings are referred to as prompting nowadays and seem to be even more verified as the number of parameters grows and are easily observed with more than 7 Billion parameters (Touvron et al., 2023a; Chowdhery et al., 2022). In this article, we refer to prompting as every modification of the input to condition the prediction.

However, large pre-trained LLMs can adapt to new tasks, only giving additional context to their input. This phenomenon considerably alleviates the need for computational resources to perform a

new task. In context-learning (Brown et al., 2020) is one of the few-shot frameworks that does not need heavy adaptation, as it simply relies on prepending input with demonstration examples. The zero-shot setting can be considered for larger language models with billions of parameters by only describing the task in natural language (Wei et al., 2022). Those two settings, referring to prompting approaches, are even more effective with the increase in the pre-trained model size (Touvron et al., 2023b; Raffel et al., 2020; Brown et al., 2020; Workshop et al., 2023). In this context, numerous studies have emerged on the identification of "good" prompts characteristics (Shin et al., 2020), or automatic prompt selection (Kojima et al., 2022; Wei et al., 2022).

In this work, we do not explore more complex prompt methodology such as few-shot learning (Brown et al., 2020) or chain-of-thought (Kojima et al., 2022). For the former, the reason is that the choice and number of examples induce too much freedom and complexity. For the latter, Chain-of-thought has shown the best results in closed models with a very high number of parameters.

Moreover, only some works attempt to explain why and how the prompts are now such a powerful tool. Even fewer studies have studied the intrinsic effect of prompts on data representation. . To the best of our knowledge, no works have studied the impact of zero-shot prompting approaches on the geometry of latent representation

Stepping slightly aside from the prompting paradigm, researchers conceived tools to study latent representations of texts. Even though Deep Neural Models still are black boxes, the explanation methods gave helpful information on the latent spaces (Aghajanyan et al., 2021; He et al., 2022), the mutual influence of tokens (Kletz et al., 2023) or the layer-wise similarity of different training techniques or models Kornblith et al. (2019). As far as we know, these methods have not been investigated to study LLMs representation leveraging prompting approaches, leading to the following question: How do variations in prompts influence the structure and distribution of vector representations in large language models?

To answer the question, we propose to divide our study into 2 directions: First, do prompts modify the intrinsic dimensionality of representations? (**RQ1.**) We believe this is an important question since few works have shown that isotropy (*i.e* the variance of a vector family is uniformly distributed across all dimensions) correlates with improved performance of embedding models (Ethayarajh, 2019; Cai et al., 2021; Liang et al., 2021; Rudman et al., 2022; Xiao et al., 2023). More generally, a better understanding of this question will enlighten us on how dimensions of LLMs are used to process queries. Second, can prompts be regrouped based on their influence on model performance and vector representation using clustering methods? (**RQ2.**) This second question adopts a more general and practical perspective. Indeed researchers have previously identified clusters and structures within deep neural representations (Phang, 2021; Cai, 2021) and established a link between these and knowledge detection. Our objective is to establish a direct correlation between the prompts, latent representations, and the model performance.

To answer these questions, we propose to verify the two following hypotheses: Prompt significantly modifies the geometry of the latent space concentration (HP1) and can be observed through the intrinsic space dimensions. The geometrical characteristics are sufficiently discriminating to facilitate the grouping or separation of prompts and comprehend how the model processes them (HP2).

The contributions of the paper are the following :

- We show that prompts modify the vector distribution on the latent space in a non-negligible way, analyzing the End-Of-Sentence representation of prompted examples on LLMs.
- LLMs do not group prompts in an expected way, meaning they focus on more geometrical features than only semantic characteristics of prompts

The remainder of the article is organized as follows. We first give context on related works in section 2. We describe our methodology in section 3. section 4 exposes the modalities and configuration of our experimentation. Then, analyses of the latent space geometry are given in section 5. We finally conclude and discuss future works in the section 6.

## 2 RELATED WORKS

**Large Language Models & Prompting** Nowadays, most state-of-the-art language models are based on the transformer architecture proposed by Vaswani et al. (2017). These architectures can be

easily adapted to various downstream tasks, such as text classification (Devlin et al., 2019) or text generation (Radford et al., 2018). For generative tasks, most transformer architectures are based on the decoder-only variant trained on an auto-regressive task (such as next token prediction) (Brown et al., 2020; Chowdhery et al., 2022; Touvron et al., 2023a; Workshop et al., 2023). Those later, when trained with billions of parameters and examples, are particularly well suited for prompting approaches Reynolds & McDonell (2021); Liu et al. (2023); Sun et al. (2023). Available decoders generally come in different flavors, raw pre-trained models, and the same model fine-tuned on instructions, namely"Instruction Tuning" (IT). IT is a more precise and efficient method to propose models that are specially designed to be efficient with prompt strategies Wei et al. (2022); Ouyang et al. (2022). During the fine-tuning, the model is fed with different queries describing the downstream task, such as "Given the text [text], could you answer the question [question]:". Recent models as Bloomz (Muennighoff et al., 2023) the IT version of bloom (Workshop et al., 2023), LLaMa (Touvron et al., 2023a;b; AI@Meta, 2024), Gemma (Gemma Team et al., 2024), Phi (Gunasekar et al., 2023; Li et al., 2023; Abdin et al., 2024) all come with and IT adapted version.

**Latent space analysis** The common acceptance of NLP axioms is based on the distributional hypothesis Firth (1957), meaning semantically close words should appear in similar contexts (texts). Thus, semantically close words should have close vector representations Mikolov et al. (2013). First, the influence of context on tokens is inherent to the pre-training tasks (Thomas et al., 2020). Second, latent spaces used to embed words are typically high-dimensional $\mathbb{R}$-vector space.

Modern model architectures extensively use the aforementioned hypothesis and its realization as contextual embedding (Mikolov et al., 2013; Devlin et al., 2019; Radford et al., 2018). The representation of a token (*e.g.* a subword) is computed with the other part of the text. Such a construction allows the extraction of the meaning from the context without specifying rules or a frozen definition (Mikolov et al., 2013). However, this comes with a disadvantage: different models produce different and non-comparable representations(Kornblith et al., 2019). The final representation highly depends indeed on the model architecture, the pre-training task (Radford et al., 2018), and the pre-training dataset(Zhou et al., 2023).
This leads to a paradox: the very same piece of text has different representations. Thus, it is merely impossible to compare them or understand the features or properties encoded in the representation Kornblith et al. (2019).

However, different signals have been explored to correlate them to the performances. Notably, studying the latent representation distribution on the latent space with metrics such as the cosine similarity Xiao et al. (2023) established a connection with model performance. However, the latter only captures the similarity of vectors and not properties on the global representation space such as effective dimension.
The isocore (Rudman et al., 2022) has been proposed to address those issues, having properties that allow robust study of the latent space based on uniformity of variance. Contrary to the explained variance, the score is computed across all dimensions, thus alleviating the need to fix an empirical threshold.
Interestingly, Ethayarajh (2019) have stated that during the training steps of LLMs, the isotropy tends to increase in latent representation and hence performances. Later, Cai et al. (2021) stated that "perfect isotropy that could explain the large model capacity", supporting the hypothesis that isotropy could be related to model performances. Furthermore, the authors stated that isotropy could be used to detect clusters and low-dimensional manifolds in latent spaces.

Still in geometrical approaches, to better understand LLM capacities, Phang et al. (2021) noticed that strong similarities occur between the first layer block and last layer blocks, suggesting that fine-tuned models in the later layer contribute marginally to the decision. These previous works confirm that isotropy deserves to be studied, as it seems to have a direct impact on performance or can provide information about the model's capacities.

## 3 METHODOLOGY

This section presents the methodologies that have been developed for this study. In order to investigate the influence of prompts throughout the construction of the output representation, it is first necessary to extract the various inner representations of the prompts (hidden states). Subsequently,

two algorithms are presented which have been employed to measure the distribution of a vector family in its extrinsic space. Subsequently, we put forth a methodology for grouping prompts through the utilisation of clustering algorithms.

**Hidden states extraction**    The initial step is to extract the hidden states. The hidden states are vector representations that capture contextual information within the Transformers framework. Therefore, the contextual representation vary depending on prompts, datasets and pre-training corpus. The following experimental setup is proposed for the purpose of studying the impact of those changes on the latent representation.

Let $\mathcal{M}, \mathcal{D}, \mathcal{P_D}$ respectively be a pre-trained model, a dataset, a prompt set adapted to $\mathcal{D}$.

For each example $e \in \mathcal{D}$, only the last generated representation is able to capture all contextual information, Therefore only the last token representation associated to the EOS (End Of Sentence) token in the language modeling head is extracted. Its representation is denoted $e_l^p$, at each layers $l \in \mathcal{M}$ and for each prompt $p \in \mathcal{P_D}$. This enables an analysis of both the representations themselves and their evolution.

**Dimensionnality & Isotropy**    We hypothesize that the prompt quality (related to task performance) directly influences the use of latent vector space. Subsequently, we propose to study two algorithms measuring the use of dimensions on inner representation space. The initial step is to undertake a Principal Component Analysis (PCA) and to make a comparative assessment of the prompts on the basis of the variance explained ratio of the first few principal components. PCA provides a highly interpretable and straightforward method of dimension reduction based on the variance in a point cloud. However, PCA get some limitations, it does not provide an absolute measure of the number of dimensions employed and exhibits instability in high-dimensional settings. In order to refine the result, the Isoscore is employed for the measurement of the effective utilisation of dimensions. IsoScore is a metric based on the PCA algorithm, which is designed to indicate the proportion of dimensions utilized by a given vector set. As outlined by Rudman et al. , the IsoScore has the following advantages: it is mean agnostic, rotation invariant and has stable scaling, which makes it an appropriate tool for comparison. PCA is used to ground analyses obtained with IsoScore. The main motivation is to compare how the latent vector space is filled with vector representations. An isoscore of 1 means that variance is homogeneous along all dimensions whereas an isoscore of 0 would mean that variance is zero.
For both methods, we compare the quantities for models and prompts and how they vary through the layers.

**Clustering**    The second hypothesis (H2) posits that point clouds exhibit discriminating characteristics, thereby enabling the generation of grouping prompts. The primary objective is to ascertain whether the geometrical characteristics are sufficiently discriminating to facilitate the grouping or separation of prompts and to comprehend the manner in which the model processes them. In order to group representations, it is proposed that clustering algorithms be used, with the number of prompts serving as the sole supervisory signal. The prompts are then to be grouped on the basis of their latent representation, a prompt is associated to a cluster if the majority of examples of the prompt belong to it. This methods is repeated for each layers of the model. Given a clustering method, $\mathbf{cluster}_k$, with $k \in \mathbb{N}^*$ the number of prompts, the prompted examples are grouped layer-wise. The quality of the clustering is evaluated using the random index score (RIS). The RIS assesses the extent to which a pair of examples, presumed to belong to the same cluster, are correctly labelled. A high RIS indicates that the clusters align with the prompts in our setup. Therefore, for a given prompt $p$ and layer $l$, all the $e_l^p \in E_l^p$ (*i.e* the latent representation generated by layer $l$ conditioned by prompt $p$) belong to the same cluster

$$c = \mathrm{argmax}_k(\mathrm{card}(\{\mathbf{cluster}(E_l^p) = k\})),$$

with $E_l^p$ the set of all representations produced by layer $l$ on the examples prompted with $p$. We get $k' \leq k$ new labels.

When the value of k is equal to itself, this signifies that the clusters are identical. However, in instances where k' is less than k, we obtain superclusters, which are clusters of clusters, that allow us to characterise similar groups of prompts.

The super-clusters show how prompts are grouped into similar clusters, thereby enabling us to examine their similarities. Furthermore, monitoring the numbers across layers provides an additional measure of representation diversity resulting from prompts.

# 4 EXPERIMENTAL PROTOCOL

This section provides a detailed description of the experimental protocol. This section begins with an overview of the models and datasets used in the experiments. Then, algorithms and strategies employed are precisely described for the prompting and classification of textual data using generative models. We subsequently illustrate the application of the aforementioned methodologies to the analysis of representations, as detailed in Section 3.

## 4.1 MODELS

It is also noteworthy that other models provide minimal information regarding their pre-training data, which increases the likelihood of data contamination. This study focuses on four state-of-the-art model families: Phi, Gemma, and Zephyr. The fourth family is Bloomz. However, due to data contamination[1], it is only used for prototyping purposes. It is also noteworthy that other models provide minimal information regarding their pre-training data, which increases the likelihood of data contamination.

**Gemma** (Gemma Team et al., 2024) is a family of model released by Google company. The team released 2B and 7B parameters versions, respectively, comprising 18 and 28 layers of respective hidden dimensions 2048 and 3072, both with an instruction-tuned variant. The Gemma team made extensive work on the architecture using several state-of-the-art modifications to the original transformer.

**Phi** (Gunasekar et al., 2023; Li et al., 2023; Abdin et al., 2024) is a family of models released by Microsoft. The latest version is Phi 3(Abdin et al., 2024). It is a 3.8B parameters decoder-only model. Phi 3 mini is composed of 32 layers with a hidden dimension of 3072. We only consider the instruction variant with a context length of 4k. Its main characteristic is that it was trained on textbook data for the first versions, and the latest was additionally trained on synthetic data.

**BloomZ** (Muennighoff et al., 2023) is the instruction tuned variant of Bloom (Workshop et al., 2023). It ranges from 560M (24 layers of size 1024) to 176B (70 layers of size 14 336). With the Zephyr family, it is the only model fully opened, and on which we have access to information regarding the training data.

**StableLM - Zephyr** is an IT variant of stableLM. It focuses on Data Preference Optimisation and gives transparent information on the pre-training and instruction tuning Data. We use the stableLM-Zephyr 3B, which has 32 layers with a hidden dimension of 2560. And the stableLM2-Zephyr 1.6B, which is a more recent version with 24 layers of hidden dimension 2048.

## 4.2 DATASETS

As stated in the Introduction (Section 1), this study focuses on binary classification tasks to enhance control over the generation process. A prototypical binary classification task is sentiment analysis, wherein two labels —positive and negative— are to be predicted. Three datasets of different sizes were selected for analysis. The composition and topics of the datasets are presented in Table 1.

The test split is used for all datasets to minimise data given no training is needed.

**Rotten Tomatoes** (Pang & Lee, 2005) is a movie review dataset containing $5,331$ positive and $5,331$ negative processed sentences from Rotten Tomatoes movie reviews. The test split contains 1064 balanced examples.

**IMDB** (Maas et al., 2011) is a dataset for binary sentiment classification from the IMDB website. The test set contains $25,000$ examples for each label.

---

[1]The Promptsource library (Bach et al., 2022) was used to produce the instruction dataset Muennighoff et al. (2023) of Bloomz.

| Dataset | positive/negative | Topic |
|---|---|---|
| Rotten Tomatoes | 532/532 | Movie Review |
| IMDB | 12500/12500 | Movie Review |
| YELP | 19000/19000 | Tourism Review |

Table 1: Summary of the dataset characteristics, with *positive/negative* standing for the number of positive and negative examples we experiment with.

**YELP** (Zhang et al., 2015) is a dataset for binary sentiment classification. It comprises a set of $38,000$ balanced reviews for testing.

## 4.3 EXPERIMENTS

Experiments are conducted across all instruction variants of models of each aforementioned family. We describe, as follow, all experiments that have been conducted.

**Prompting**  The prompting methodology is based on the Promptsource library Bach et al. (2022) and its default prompt templates. In order to ensure at least some category of prompt we can isolate and control, we take the default templates and duplicate them with minor modifications to isolate those characteristics. All prompts consisting of templates containing the instructions/query and the context (the text to classify), with specific labels to predict, for instance :

*[context] The sentiment expressed for the movie is [label]*

Where [context] is the text to classify and [label] the word to predict (in this example, "positive" or "negative" is expected), the label is not provided in the input of the model (unless otherwise specified). The significant advantage of Promptsource is that it comes with numerous recognized datasets and simplifies the experimental protocol. It should be acknowledged that Promptsource is the tool employed to refine BloomZ and mitigate potential data contamination in other models, given the dearth of information regarding pre-training datasets, particularly in the case of Gemma and Phi.

**Classification methodology**  Classification with generative models is more handy than with encoder models where a standard classifier is trained on last latent representation. Moreover, since we focus on the zero-shot framework, we prefer not to train a classifier head on top of the model. To avoid too much burden, we constraint the model to output only the wanted tokens by setting logits for other indexes to $-\infty$. This procedure allows the alignment of different models' outputs. Then, we follow Wei et al. (2022) and compare the ranking in the output distribution between labels. For the binary case, it can be written :

$$y = \operatorname{argmax}_{l_i}(\mathrm{P}(F(x, \theta) = l_i))$$

with $F(\cdot, \theta)$ the LLM, $x$ the input example, $y$ the retained prediction and $l_{\{1,2\}}$ the labels to predict. If the labels are tokenized in multiple tokens, we take the product of the probabilities to produce a sequence probability and eventually compare them.

## 5 RESULTS

Results are grouped and discussed into our two research questions we give more general analyses at the end of this section. First analyse results of isoscore are discussed to support the HP1 . Second an exploration of the possibility to grouping prompts using their inner representations.

## 5.1 ISOTROPY

To measure the isotropy we compute the IsoScore prompt and model wise. We analyze how prompts modify the vector distribution on their space accross layers. Since IsoScore is a recent algorithms aiming to make use of the PCA, we conducted similar analyses with PCA we report in the Annex A.

In the figure 5.1 we report the mean isoscore for each model and dataset. Each plot present the average isoscore along each layer, the figure also represent the standard deviation for each model across prompts as error bars. This let us draw the following general analyses.

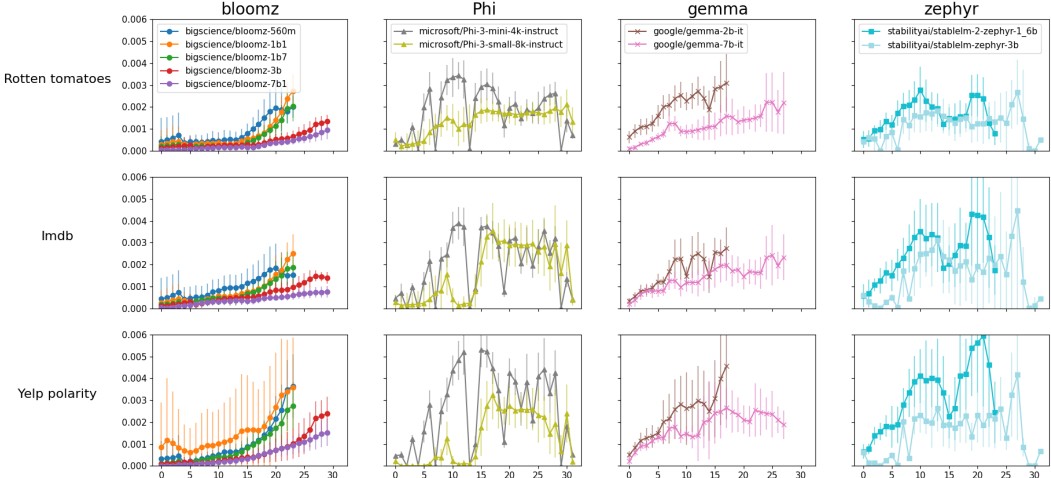

Figure 1: Mean IsoScore through layers per dataset (rows) and model families (columns).

First, we notice that the effective use of dimension is very low, ranging from $0$ to $0.006$, meaning that at maximum $0.6\%$ of dimension are sufficient to differentiate the different representations of the "EOS" tokens (among examples or prompts). This behavior is expected since we only analyze the `EOS` representation; this leads to an increased similarity between the vectors we compare. Moreover, high similarities of auto-regressive model representations have already been noticed (Phang et al., 2021; Cai et al., 2021).

Second, those quantities show a common behavior for most models. The use of space (represented by the curves) tends to increase through the layers. This behavior is more clearly observed for the Bloomz and Gemma families, which show a regular increase through the layer. A possible interpretation could rely on the architectures of those models that used the last representation to model the language, selecting tokens beyond all possibles. The number of possible choices and the complexity of the language modeling task could lead to exploit a larger number of dimensions. While we can observe a similar tendency for Phi and Zephyr, the trend is less apparent. Thus, it could also be due to the training step or pre-training data (since the architectures of Gemma, Phi, and Zephyr are highly similar).

Third, the evolution seems to be highly dependent on the number of layers; smaller models generally have a higher IsoScore than their larger counterpart for a given family, as seen in Figure 5.1. Moreover, Table 2 compares the mean Isoscore over the models and prompt and shows that it depends more on the former. Indeed, models with fewer layers tend to have a higher IsoScore together with a faster increase (*e.g.* comparing gemma-2b-it and gemma-7b-it provides a good example of this phenomenon).

|  | rotten tomatoes | imdb | yelp polarity |
|---|---|---|---|
| Mean IsoScore per model | 48.92% | 51.74% | 55.01% |
| Mean IsoScore per prompt | 74.28% | 71.18% | 78.79% |

Table 2: Mean standard deviation computed over the models and the prompts for each DataSet

Fourth, Figure5.1 shows the mean isotropy (IsoScore) of the different prompts for a subset of models, namely Bloomz 1b7, Gemma 2B and StableLM 2 Zephyr 1.6B on IMDB colored by their accuracy scores. The choice of these model is motivated because of their similar size. Though we cannot link the IsoScore to the performance of the model and prompt with this Figure, we notice that efficient prompts show similar evolution. Using this figure,the analysis of dimensionality shows differences between prompts for most layers which translates the importance of the way prompts are

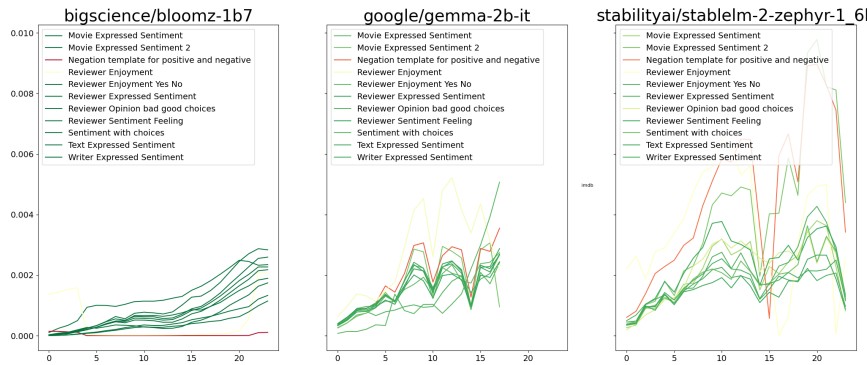

Figure 2: Evolution of the IsoScore through layers of different prompts on the IMDB dataset for Bloomz 1b7,Gemma 2B and StableLM 2 Zephyr 1.6B. Greener represent a higher score while redder a lower. A yellow line denotes a score close to $0.5$.

formulated even though examples and performance can be similar (Figure 2). Table 3 reports the mean standard deviation across of the IsoScore normalized by the mean IsoScore. For every model, the percentage is not negligible showing a sturdy effect of the prompt on internal vector space.

This means that even though isoscore is not a relevant measure to analyze the efficiency of prompts, bad prompts tend to destabilize internal representations, yielding either too concentrated or too diffuse representation.

Moreover, the evolution of the isoscore is smoother for the Bloomz family, as seen on Figures 5.1 and 2. One possible explanation is that the Bloomz models use the prompt dataset for IT, it can be a syndrome of the pre-training knowledge. Notice that we cannot totally state on the hypothesis since Zephyr is also trained on the datasets, however, probably with different prompts.

Table 3: Mean standard deviation across layer expressed as a percentage of the mean isoscore per model for each dataset

|  | rotten tomatoes | imdb | yelp polarity |
| --- | --- | --- | --- |
| bigscience/bloomz-560m | 70.62% | 73.22% | 60.24% |
| bigscience/bloomz-1b1 | 59.13% | 61.39% | 131.36% |
| bigscience/bloomz-1b7 | 58.21% | 66.15% | 58.09% |
| bigscience/bloomz-3b | 58.07% | 42.94% | 61.04% |
| bigscience/bloomz-7b1 | 45.15% | 42.72% | 42.39% |
| google/gemma-2b-it | 29.3% | 33.53% | 49.19% |
| google/gemma-7b-it | 46.52% | 41.68% | 40.03% |
| microsoft/Phi-3-mini-4k-instruct | 26.35% | 21.14% | 21.66% |
| microsoft/Phi-3-small-8k-instruct | 42.03% | 53.19% | 47.93% |
| stabilityai/stablelm-2-zephyr-1_6b | 38.97% | 48.97% | 42.59% |
| stabilityai/stablelm-zephyr-3b | 63.72% | 84.22% | 50.61% |

With those experiments and the results obtained, we can now provide answers to **RQ1**. The prompts do influence the way representations are distributed on the vector space. However, there is no apparent monotonic correlation or relation between isotropy and model performances. Nevertheless, according to the figure 2 bad performance seems to be correlated with extreme isotropy.

## 5.2 CLUSTERS

The use of clustering algorithms along with the majority vote shows interesting results. The KMeans algorithm shows a good agreement of the clusters using a Random Index Score (RIS). Second, after the majority vote, the number of resulting clusters is often lower (Figure 3. This means that clustering tends to focus on other characteristics than the prompt itself. Moreover, evaluating the majority

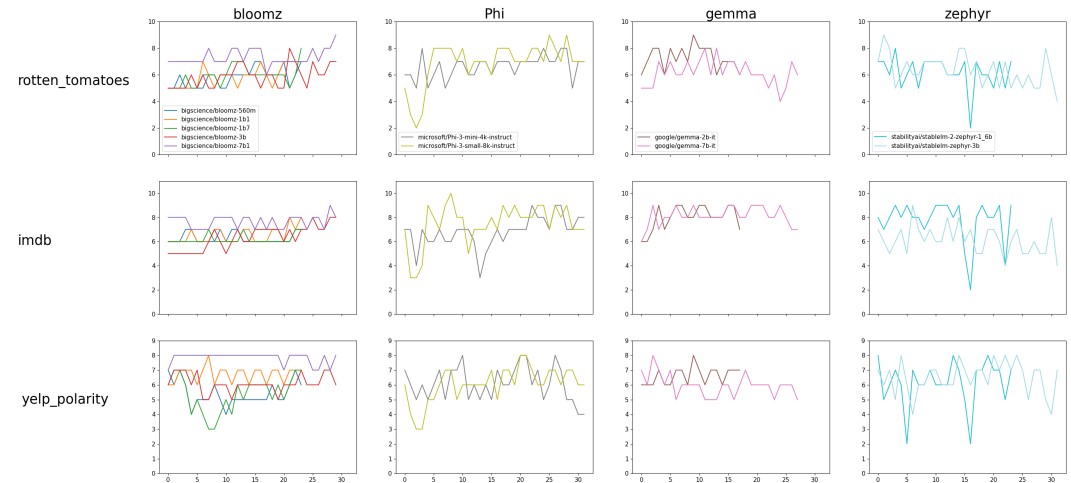

Figure 3: Number of prompts obtained after a majority vote layer-wise per dataset (rows) and model families (columns).

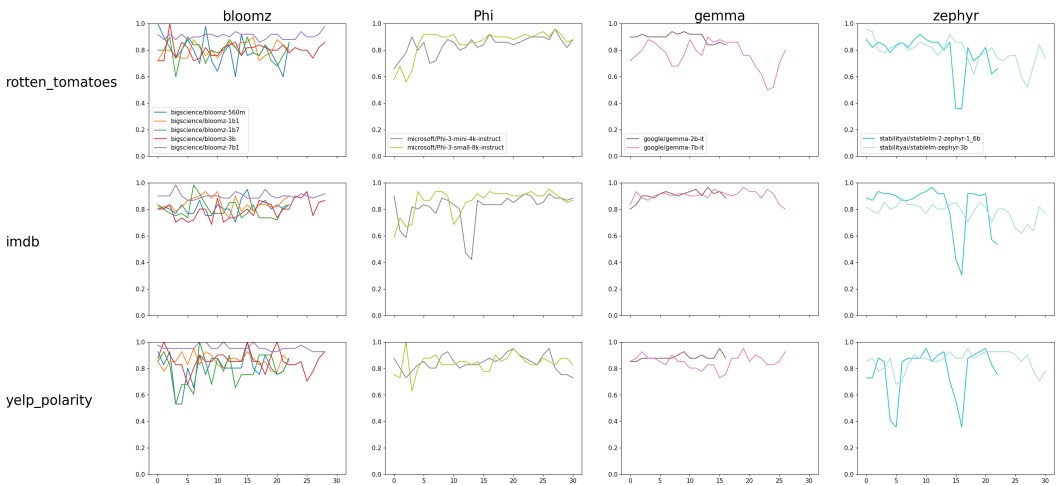

Figure 4: Evolution of the RIS after majority vote on consecutive layers.

vote on consecutive layers (Figure 4) also shows a good RIS, meaning that prompts are grouped consistently through the layers. This shows that there are features in the vector representation that go beyond the prompts and that can be used to regroup the prompts.

Figure 3 shows the number of majority clusters through the layers and denotes the diversity produced by the different prompts.

Table 4 shows the number of times the prompts `"Movie Expressed Sentiment"` (M0), `"Movie Expressed Sentiment 2"` (M1), `"Text Expressed Sentiment"` (S0), `"Writer Expressed Sentiment"` (S1) were grouped together. This table shows unexpected results as it seemed reasonable to group the first two prompts together and the last one together. However `"Movie Expressed Sentiment 2"` and `"Text Expressed Sentiment"` are grouped more often (∼20% of the time). We produce the full table in the appendix.

This allows us give answers to **RQ2**. First, a simple KMean clustering is able to distinguish prompts with a good RIS and group some of the prompts with respect to other characteristics. Second, after a majority vote the clustering is quite stable across layers, meaning that the first quality evoked is stable across the model. Finally, diving into the grouped prompts gives a counter-intuitive clustering,

Table 4: Exemple of the number of time four prompts (`"Movie Expressed Sentiment"` (M0),`"Movie Expressed Sentiment 2"` (M1),`"Text Expressed Sentiment"` (S0),`"Writer Expressed Sentiment"` (S1)) were grouped after a majority vote on IMDB on all models and layers

|    | M0     | M1     | S0     | S1     |
|----|--------|--------|--------|--------|
| M0 | 100.0% | 6.71%  | 5.7%   | 7.72%  |
| M1 | 6.71%  | 100.0% | 12.75% | 20.81% |
| S0 | 5.7%   | 12.75% | 100.0% | 13.09% |
| S1 | 7.72%  | 20.81% | 13.09% | 100.0% |

showing that the geometrical features used by the clustering algorithm weakly correspond to the semantic attributes of the prompts.

## 6 CONCLUSIONS

This study investigated the correlations between diverse prompts and the latent representation in large language models (LLMs). Our research employs two distinct approaches. The first is a study of vector distributions within the latent space at the layer level. The second is investigating the possibility for grouping prompts based solely on the latent representations they produce.

The distribution of the vectors shows differences through prompts for each model and each dataset. These differences depend on two aspects. First, different prompts produce different distributions at each layer, indicating their importance at each step of the LLMs. This means that the models process prompts differently up to the prediction stage. Second, the study of the isotropy evolution of the latent representation shows behavior that depends on the model family rather than on prompts and datasets. This shows that architectures, pre-training data, and training paradigms leave detectable traces of how models process their inputs.

The possibility of grouping prompts using only the latent representations shows that vector representations contain specific properties that depend on models and datasets. This result is counterintuitive as we would expect prompts that are close in natural language to be treated similarly and thus grouped in the same cluster. However, each model group prompts differently, and the resulting clusters are sometimes unexpected. A reasonable interpretation is that the pre-trained knowledge leveraged by models is very sensitive to the input form and its modification.

These two findings lead to the following statement: the internal representation of models is highly dependent on small changes in the input, whether from a distributional or a structural point of view. While this may seem like a reasonable and expected statement, it highlights the importance of studying the deeper processing of inputs in order to understand how a model works and why prompts can lead to correct or incorrect predictions.

In future work, we plan to propose more robust and novel methods to accurately identify the defining features we have identified. A major objective should be to link specific geometric features of both models to linguistic ones.

ACKNOWLEDGMENTS

This work was granted access to the HPC resources of XXXXX under the allocation 20XX made by YYYY

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

## A  APPENDIX

ISOSCORE

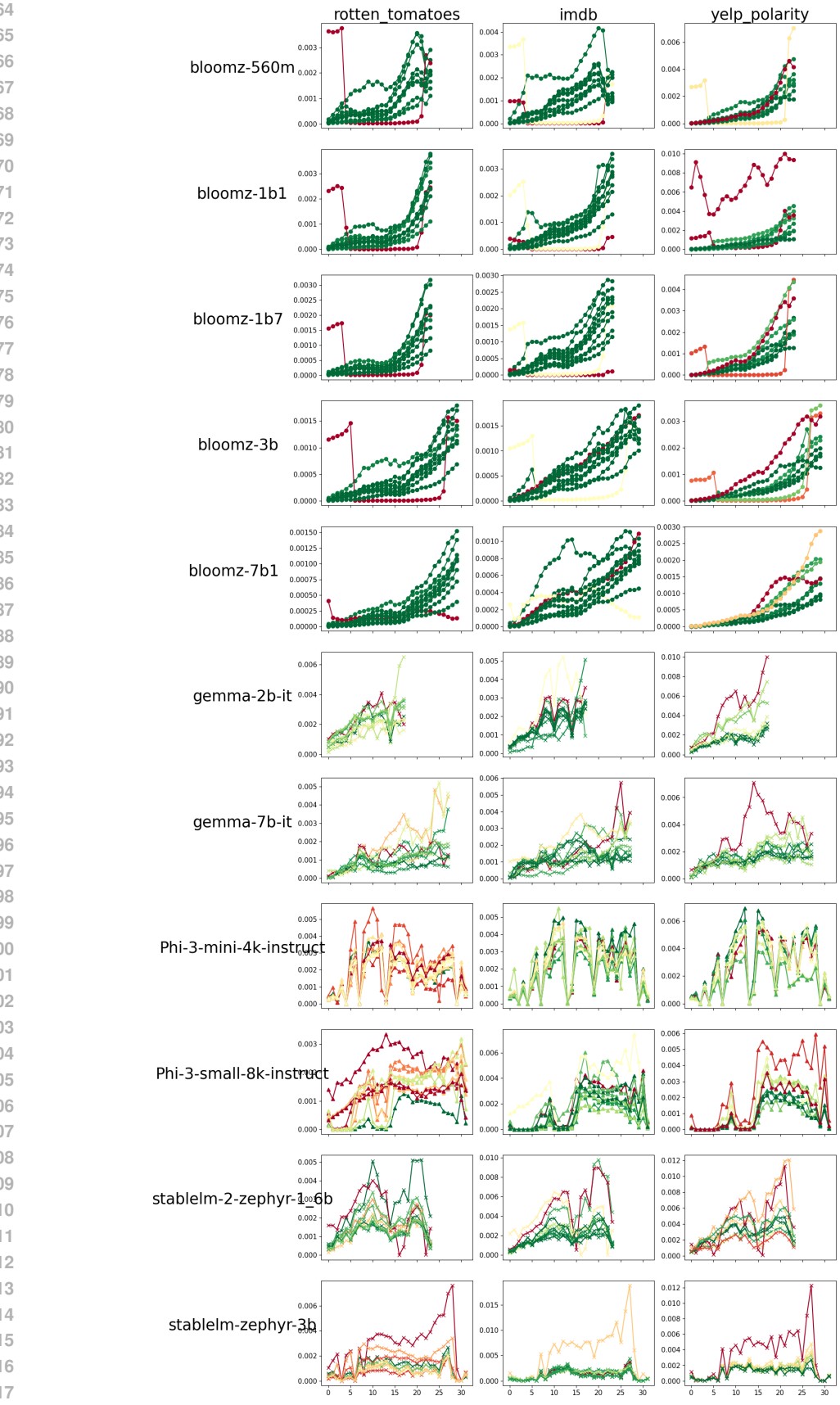

Figure 5: Isoscore per prompts and models for each dataset colored by Accuracy score

VARIANCE EXPLAINED BY PCA DIMENSION

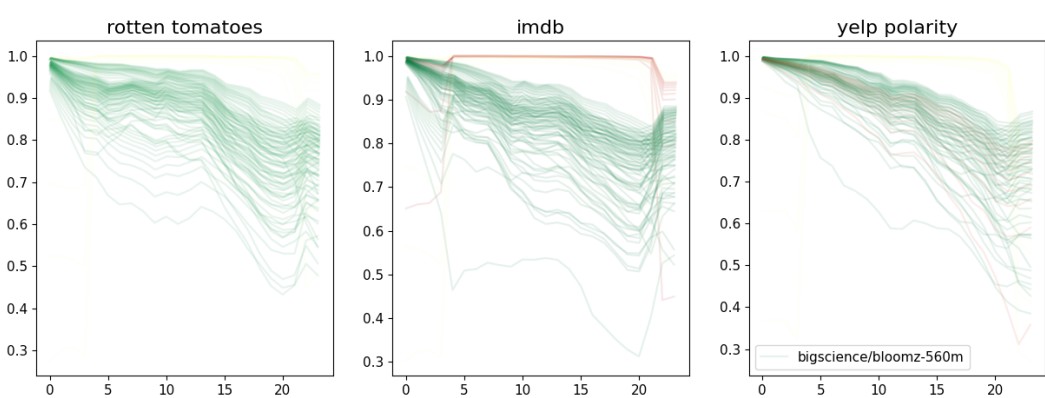

Figure 6: 10 first dimensions var explained by PCA bigscience bloomz-560m

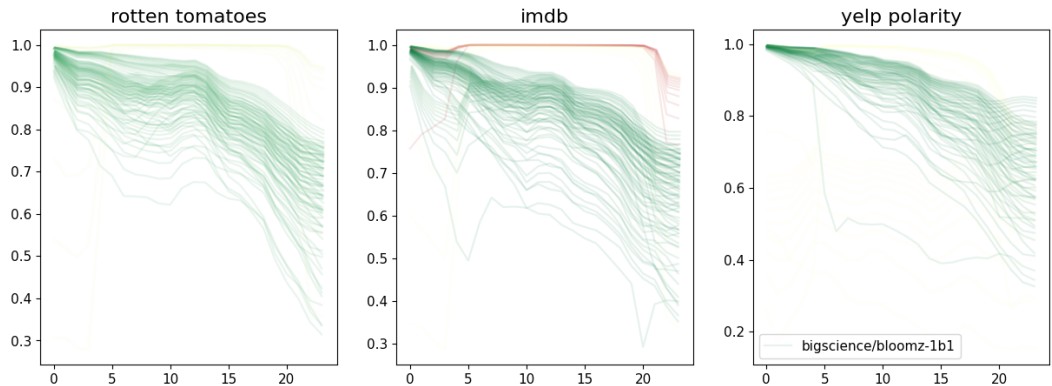

Figure 7: 10 first dimensions var explained by PCA bigscience bloomz-1b1

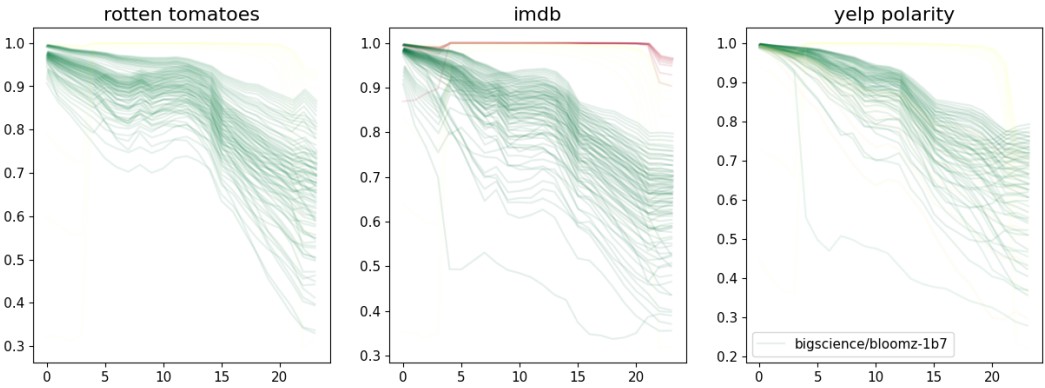

Figure 8: 10 first dimensions var explained by PCA bigscience bloomz-1b7

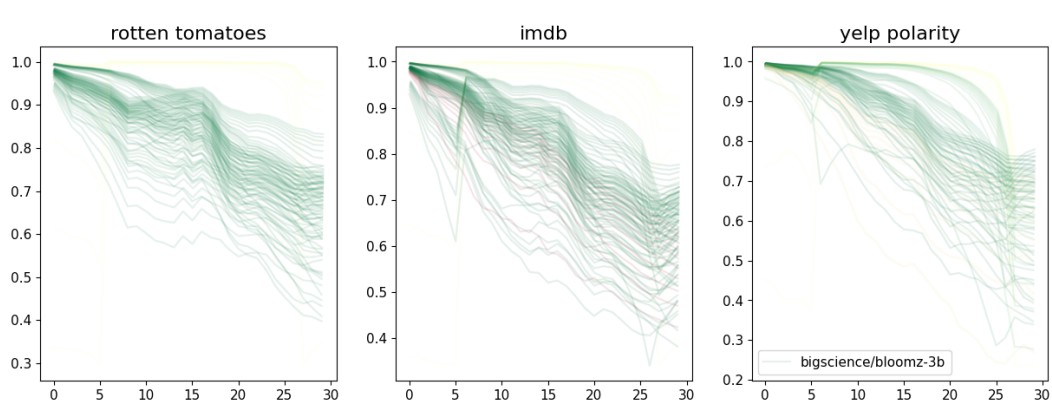

Figure 9: 10 first dimensions var explained by PCA bigscience bloomz-3b

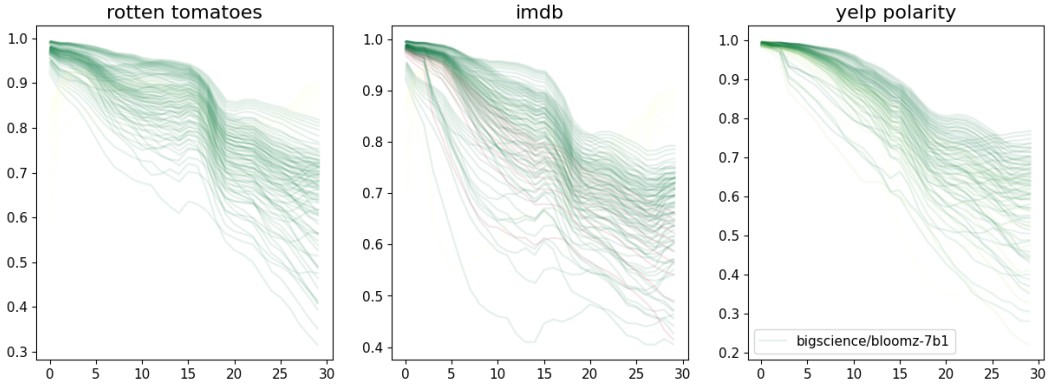

Figure 10: 10 first dimensions var explained by PCA bigscience bloomz-7b1

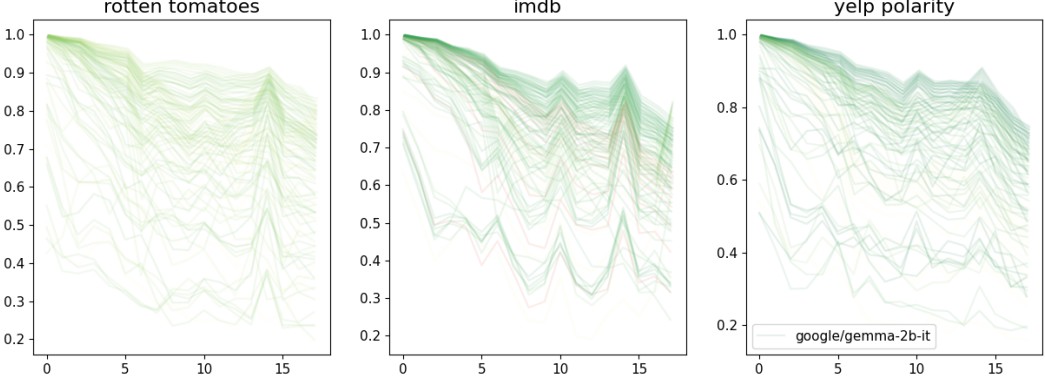

Figure 11: 10 first dimensions var explained by PCA google gemma-2b-it

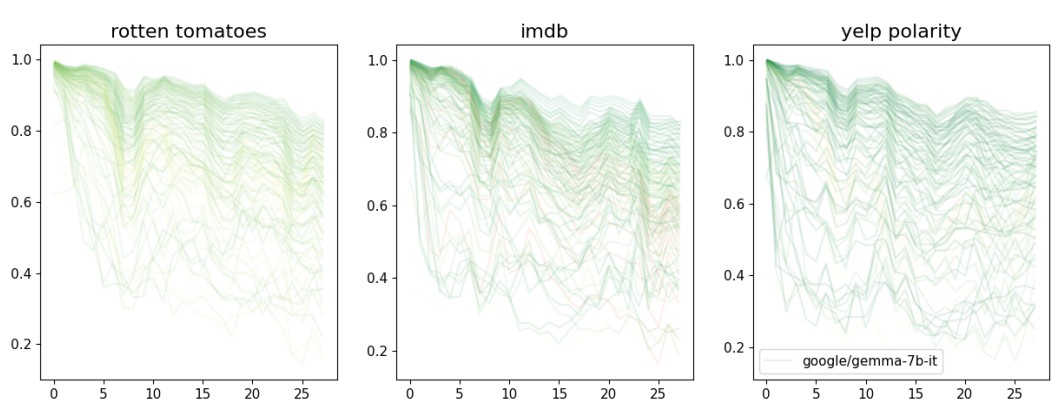

Figure 12: 10 first dimensions var explained by PCA google gemma-7b-it

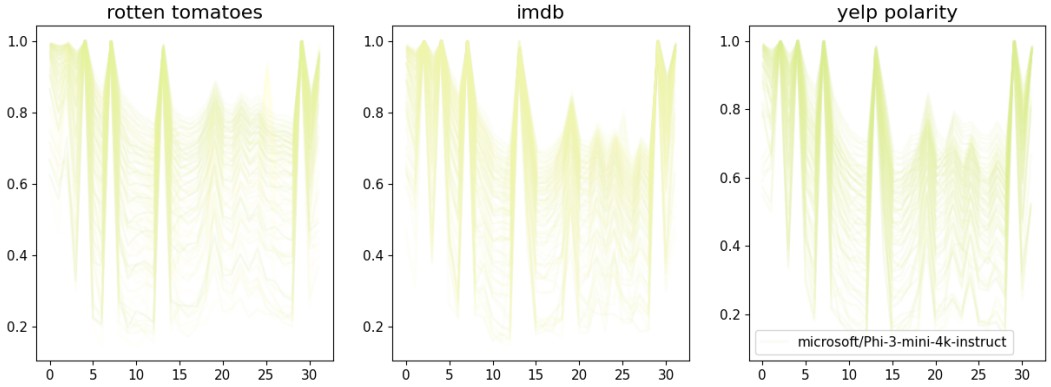

Figure 13: 10 first dimensions var explained by PCA microsoft Phi-3-mini-4k-instruct

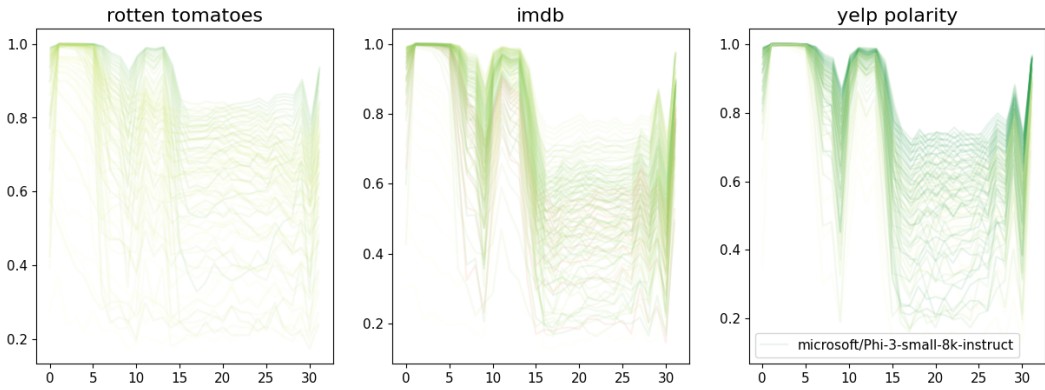

Figure 14: 10 first dimensions var explained by PCA microsoft Phi-3-small-8k-instruct

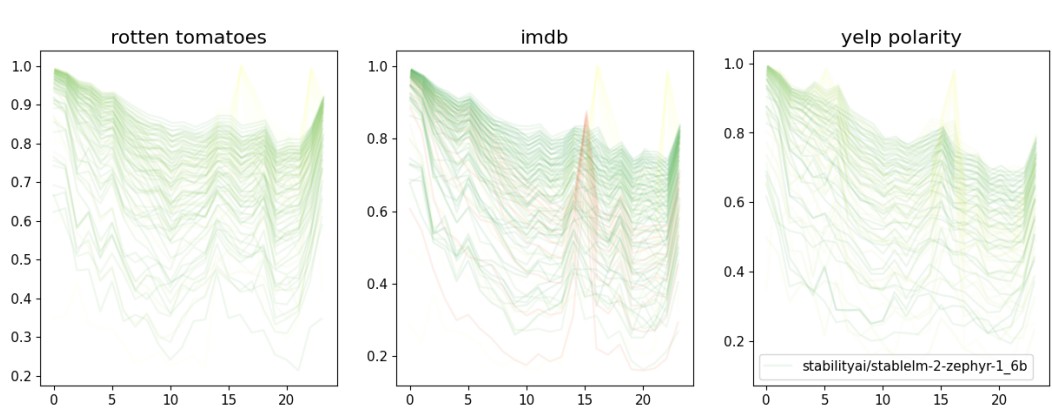

Figure 15: 10 first dimensions var explained by PCA stabilityai stablelm-2-zephyr-1$_6b$

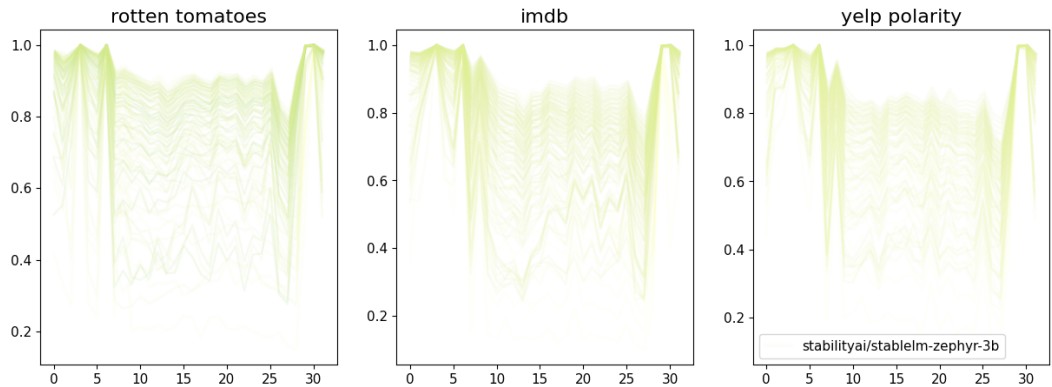

Figure 16: 10 first dimensions var explained by PCA stabilityai stablelm-zephyr-3b

MAJORITY VOTES ON PROMPTS

Table 5: Number of vote on rotten tomatoes on all models and layers for the following prompts: Movie Expressed Sentiment (MES), Movie Expressed Sentiment 2 (MES 2), Reviewer Enjoyment (RE), Reviewer Enjoyment Yes No (REYN), Reviewer Expressed Sentiment (RES), Reviewer Opinion bad good choices (ROBGY), Reviewer Sentiment Feeling (RSF), Sentiment with choices (SC), Text Expressed Sentiment (TES), Writer Expressed Sentiment (WES)

| | MES | MES 2 | RE | REYN | RES | ROBGC | RSF | SC | TES | WES |
|---|---|---|---|---|---|---|---|---|---|---|
| MES | 100.0% | 4.7% | 7.05% | 4.7% | 4.36% | 9.4% | 9.73% | 9.73% | 7.05% | 10.07% |
| MES 2 | 4.7% | 100.0% | 6.71% | 11.07% | 9.73% | 23.15% | 17.45% | 22.15% | 20.81% | 18.12% |
| RE | 7.05% | 6.71% | 100.0% | 6.38% | 9.06% | 14.09% | 10.74% | 17.11% | 12.75% | 12.42% |
| REYN | 4.7% | 11.07% | 6.38% | 100.0% | 5.7% | 10.74% | 14.09% | 14.77% | 13.09% | 12.75% |
| RES | 4.36% | 9.73% | 9.06% | 5.7% | 100.0% | 17.11% | 13.42% | 15.44% | 17.11% | 16.44% |
| ROBGC | 9.4% | 23.15% | 14.09% | 10.74% | 17.11% | 100.0% | 14.77% | 16.78% | 19.8% | 20.13% |
| RSF | 9.73% | 17.45% | 10.74% | 14.09% | 13.42% | 14.77% | 100.0% | 14.77% | 16.78% | 14.43% |
| SWC | 9.73% | 22.15% | 17.11% | 14.77% | 15.44% | 16.78% | 14.77% | 100.0% | 19.46% | 16.44% |
| TES | 7.05% | 20.81% | 12.75% | 13.09% | 17.11% | 19.8% | 16.78% | 19.46% | 100.0% | 17.11% |
| WES | 10.07% | 18.12% | 12.42% | 12.75% | 16.44% | 20.13% | 14.43% | 16.44% | 17.11% | 100.0% |

Table 6: Number of vote on IMDB on all models and layers for the following prompts: Movie Expressed Sentiment (MES), Movie Expressed Sentiment 2 (MES 2), Negation template for positive and negative (NTPN), Reviewer Enjoyment (RE), Reviewer Enjoyment Yes No (REYN), Reviewer Expressed Sentiment (RES), Reviewer Opinion bad good choices (ROBGC), Reviewer Sentiment Feeling (RSF), Reviewer Sentiment with choices (SC), Text Expressed Sentiment (TES), Writer Expressed Sentiment (WES)

|        | MES    | MES 2  | NTPN   | RE     | REYN   | RES    | ROBGC  | RSF    | SC     | TES    | WES    |
|--------|--------|--------|--------|--------|--------|--------|--------|--------|--------|--------|--------|
| MES    | 100.0% | 6.71%  | 8.39%  | 5.03%  | 4.36%  | 9.73%  | 9.06%  | 5.37%  | 7.38%  | 5.7%   | 7.72%  |
| MES 2  | 6.71%  | 100.0% | 15.77% | 12.08% | 8.39%  | 24.83% | 18.79% | 15.77% | 16.11% | 12.75% | 20.81% |
| NTPN   | 8.39%  | 15.77% | 100.0% | 7.72%  | 8.72%  | 14.77% | 13.42% | 7.05%  | 10.4%  | 6.71%  | 13.42% |
| RE     | 5.03%  | 12.08% | 7.72%  | 100.0% | 6.38%  | 14.09% | 12.08% | 10.74% | 12.75% | 14.77% | 14.43% |
| REYN   | 4.36%  | 8.39%  | 8.72%  | 6.38%  | 100.0% | 9.73%  | 9.06%  | 10.07% | 14.43% | 8.72%  | 16.44% |
| RES    | 9.73%  | 24.83% | 14.77% | 14.09% | 9.73%  | 100.0% | 14.77% | 10.4%  | 13.09% | 12.75% | 16.78% |
| ROBGC  | 9.06%  | 18.79% | 13.42% | 12.08% | 9.06%  | 14.77% | 100.0% | 8.05%  | 11.41% | 12.75% | 17.11% |
| RSF    | 5.37%  | 15.77% | 7.05%  | 10.74% | 10.07% | 10.4%  | 8.05%  | 100.0% | 12.75% | 10.74% | 12.75% |
| SWC    | 7.38%  | 16.11% | 10.4%  | 12.75% | 14.43% | 13.09% | 11.41% | 12.75% | 100.0% | 15.1%  | 12.75% |
| TES    | 5.7%   | 12.75% | 6.71%  | 14.77% | 8.72%  | 12.75% | 12.75% | 10.74% | 15.1%  | 100.0% | 13.09% |
| WES    | 7.72%  | 20.81% | 13.42% | 14.43% | 16.44% | 16.78% | 17.11% | 12.75% | 12.75% | 13.09% | 100.0% |

Table 7: Number of vote on Yelp Polarity on all models and layers for prompts: come again (CA), experience good bad (EGB), format come again (DCA), format good bad (FGB), like dislike (LD), like dislike 2 (LD 2), place good bad (PGB), rating high low (RHL), regret yes or no (RYN)

| | CA | EGB | DCA | FGB | LD | LD 2 | PGB | RHL | RYN |
|---|---|---|---|---|---|---|---|---|---|
| CA | 100.0% | 5.03% | 5.7% | 5.37% | 5.03% | 7.72% | 8.05% | 4.03% | 7.05% |
| EGB | 5.03% | 100.0% | 15.77% | 20.81% | 10.4% | 20.13% | 17.45% | 20.81% | 20.13% |
| DCA | 5.7% | 15.77% | 100.0% | 14.43% | 5.7% | 13.42% | 13.76% | 13.76% | 12.42% |
| FGB | 5.37% | 20.81% | 14.43% | 100.0% | 7.05% | 16.44% | 16.11% | 14.43% | 14.09% |
| LD | 5.03% | 10.4% | 5.7% | 7.05% | 100.0% | 14.09% | 6.04% | 10.07% | 6.38% |
| like dislike 2 | 7.72% | 20.13% | 13.42% | 16.44% | 14.09% | 100.0% | 15.44% | 18.46% | 17.11% |
| PGB | 8.05% | 17.45% | 13.76% | 16.11% | 6.04% | 15.44% | 100.0% | 11.74% | 15.77% |
| RHL | 4.03% | 20.81% | 13.76% | 14.43% | 10.07% | 18.46% | 11.74% | 100.0% | 12.42% |
| RYN | 7.05% | 20.13% | 12.42% | 14.09% | 6.38% | 17.11% | 15.77% | 12.42% | 100.0% |

