# OpenReview forum: "Impact of Prompt on Latent Representations in LLMs"
_ICLR.cc/2025/Conference — Submitted to ICLR 2025_

### Official Review · Reviewer_LxPV · 2024-11-03

**Soundness:** 2
**Presentation:** 1
**Contribution:** 1
**Rating:** 3
**Confidence:** 4

**Summary:**

The paper studies how variations in the prompt change the distribution of hidden states in LLMs for zero-shot binary classification. The last hidden state representation is extracted at each layer for a variety of different prompts. The authors plot how the IsoScore of the hidden states evolves across layers. These hidden states are then clustered, based on which the authors conclude that "[the] clustering tends to focus on other characteristics than the prompt itself."

**Strengths:**

The paper studies a general and interesting question of how LLM internals (such as hidden states) evolve/change when certain parts of the input (such as the prompt) change. Further understanding of this process would be useful for several downstream applications, such as (as shown in prior work) detecting adversarial prompt injection attacks.

**Weaknesses:**

The biggest issue with the draft is the presentation of the results. There are numerous spelling, writing, and formatting errors, a subset of which I listed below. The first two paragraphs of the introduction are unnecessary background for the ICLR audience. The related work section on LLMs is not particularly related to the focus of this paper (understanding how prompts change internal model representations). That section is missing citations to LogitLens and its derivatives (e.g. https://arxiv.org/pdf/2303.08112), which also study how changes in the prompt change model internals. For example, that paper develops classifiers (based on features extracted from model internals) for detecting prompt injection. I.e., they already studied this paper's "hypothesis 2" that "The geometrical characteristics are sufficiently discriminating to facilitate the grouping or separation of prompts and comprehend how the model processes them (HP2)." The details of very relevant parts of the paper, such as a brief description of the IsoScore algorithm, are missing. Isotropy is never formally defined and the draft never explains why it is a desirable property for decoder-only models (beyond references to other work studying isotropy in the context of embedding models). The details of precisely how the IsoScore is computed are missing to the point where I'm not sure the results of the paper are reproducible.

I think minor writing and presentation issues are not disqualifying, but in this case they are pervasive and make it difficult to judge the technical aspects of the paper on merit.

Subset of writing issues:
- L16 "heuristics rules" --> "heuristic rules"
- L24 "their impact" --> "its impact"

- first two paragraph of the intro are unnecessary bg for the ICLR audience.

- L53 starts "However" but is not contradicting a previous point

- L67 what does "intrinsic" mean here?
- L67 double period
- L69 missing period.

- L74 "LLMs representations leveraging prompting" --> "LLM representations that leverage prompting"

- L101 "section 4" --> "Section 4".

- L123, 137, L286 have \citet that should be \citep

- L174 "information, Therefore" --> "information. Therefore,"

- L178 "Dimensionnality"

- L184 "PCA get some limitations"

- L214-L215 $k$ not in latex formatting

- L284 "We describe, as follow,"

- L290, L306 "for instance :", "be written :"

- L303 "we constraint the model"

- L317 "First analyse results of isoscore are discussed"

- L322 "Since IsoScore is a recent algorithms"

- L323 "accross"

- L483 "This allows us give answers"

**Questions:**

- Why are we interested in isotropy of the EOS token hidden states?

- Can we use the isoscore as a selection criterion for picking a good prompt? This possibility is hinted at in the introduction "Our objective is to establish a direct correlation between the prompts, latent representations, and the model performance." but not deeply explored. The results sound negative on this front---"we cannot link the IsoScore to the performance of the model and prompt". How does this finding relate to the cited works that show isotropy is a good property for embedding models?

---

### Official Review · Reviewer_ZnB9 · 2024-11-03

**Soundness:** 1
**Presentation:** 2
**Contribution:** 1
**Rating:** 3
**Confidence:** 4

**Summary:**

This paper investigates how prompts affect the intrinsic geometry of LLMs. The authors explore two research questions: (1) whether prompts alter the intrinsic dimensionality of representations, and (2) whether prompts can be grouped by their impact on model performance and vector representations using clustering methods.

**Strengths:**

The paper tries to address the intriguing question of how prompt representations relate to model performance, with a specific focus on how prompt formulation impacts performance robustness.

**Weaknesses:**

The paper's format is somewhat disorganized, with unexpected line breaks and misplaced punctuation. Also, some figures like figure 3 do not have illustrations, making it hard to understand.

**Questions:**

(1) What are the motivations of grouping the prompts and how the groups of prompts correlate to the focus of the model generation? Why do the authors use the last layer rather than the embedding layer for clustering the prompts?

(2) What is the mathematical formulation of the Isoscore? What does it used for in the experiments?

(3) Several studies, such as PromptRobust [1], focus on evaluating prompt robustness. How does this approach offer advantages over existing methods?

(4) The paper offers limited technical contributions and focuses narrowly on the classification task. Could the work be applied to generation task?

[1] Zhu, K., Wang, J., Zhou, J., Wang, Z., Chen, H., Wang, Y., ... & Xie, X. (2023). Promptbench: Towards evaluating the robustness of large language models on adversarial prompts. arXiv preprint arXiv:2306.04528.

---

### Official Review · Reviewer_xyjk · 2024-11-05

**Soundness:** 1
**Presentation:** 2
**Contribution:** 1
**Rating:** 3
**Confidence:** 4

**Summary:**

This paper investigates the effect of prompts on the representation of the EOS token across prompts and LLMs.

**Strengths:**

- The idea of studying the effects of prompts through the geometry of latent representations is interesting.

**Weaknesses:**

- Studying only the representation of the EOS token seems to be an oversimplification. While this representation technically can depend on all of the context/generation, it might not attend to its meaningful parts, thus failing to capture interesting patterns.
- The paper lacks clear/practical insights besides observing that the EOS token representation does depend on the prompt in some way.
- For an empirically oriented work, studying only binary sentiment classification datasets is insufficient. Hypotheses should be verified across a broader range of tasks, including open-ended generation.

Other:
- Many typos throughout the paper (missing spaces and periods, inconsistent usage of citet and citep, etc.)
- IsoScore is not defined in the paper. Only a high-level explanation is provided.
- RIS is not carefully defined - is k' the same as c? "value of k is equal to itself" is not a clear statement.

**Questions:**

See weaknesses.

---

### Meta-Review · Area_Chair_TGLc · 2024-12-20

**Metareview:**

This paper studies the effect of prompts on various geometric aspects of embeddings. This question is tackled for zero-shot binary prediction.

The overall direction of the paper is certainly interesting, and could help with building robust zero-shot methods. However, the paper doesn’t quite deliver on this vision.

As noted by the reviewers, the writing could use lots of work. The implications of the results should be further fleshed out, and comparisons to the fairly vast literature on this space should also be expanded. This iteration of the paper is below the bar, but this could be rectified in the future version.

**Additional Comments On Reviewer Discussion:**

There was no rebuttal for this paper.

---

### Decision · Program_Chairs · 2025-01-22

Reject